# A Diet Containing Rutin Ameliorates Brain Intracellular Redox Homeostasis in a Mouse Model of Alzheimer’s Disease

**DOI:** 10.3390/ijms24054863

**Published:** 2023-03-02

**Authors:** Paloma Bermejo-Bescós, Karim L. Jiménez-Aliaga, Juana Benedí, Sagrario Martín-Aragón

**Affiliations:** Department of Pharmacology, Pharmacognosy and Botany, Faculty of Pharmacy, Complutense University of Madrid, Plaza Ramón y Cajal s/n, 28040 Madrid, Spain

**Keywords:** quercetin, rutin, glutathione, APPswe, BACE1, Alzheimer’s disease

## Abstract

Quercetin has been studied extensively for its anti-Alzheimer’s disease (AD) and anti-aging effects. Our previous studies have found that quercetin and in its glycoside form, rutin, can modulate the proteasome function in neuroblastoma cells. We aimed to explore the effects of quercetin and rutin on intracellular redox homeostasis of the brain (reduced glutathione/oxidized glutathione, GSH/GSSG), its correlation with β-site APP cleaving enzyme 1 (BACE1) activity, and amyloid precursor protein (APP) expression in transgenic TgAPP mice (bearing human Swedish mutation APP transgene, APPswe). On the basis that BACE1 protein and APP processing are regulated by the ubiquitin–proteasome pathway and that supplementation with GSH protects neurons from proteasome inhibition, we investigated whether a diet containing quercetin or rutin (30 mg/kg/day, 4 weeks) diminishes several early signs of AD. Genotyping analyses of animals were carried out by PCR. In order to determine intracellular redox homeostasis, spectrofluorometric methods were adopted to quantify GSH and GSSG levels using o-phthalaldehyde and the GSH/GSSG ratio was ascertained. Levels of TBARS were determined as a marker of lipid peroxidation. Enzyme activities of SOD, CAT, GR, and GPx were determined in the cortex and hippocampus. ΒACE1 activity was measured by a secretase-specific substrate conjugated to two reporter molecules (EDANS and DABCYL). Gene expression of the main antioxidant enzymes: APP, BACE1, a Disintegrin and metalloproteinase domain-containing protein 10 (ADAM10), caspase-3, caspase-6, and inflammatory cytokines were determined by RT-PCR. First, overexpression of APPswe in TgAPP mice decreased GSH/GSSG ratio, increased malonaldehyde (MDA) levels, and, overall, decreased the main antioxidant enzyme activities in comparison to wild-type (WT) mice. Treatment of TgAPP mice with quercetin or rutin increased GSH/GSSG, diminished MDA levels, and favored the enzyme antioxidant capacity, particularly with rutin. Secondly, both APP expression and BACE1 activity were diminished with quercetin or rutin in TgAPP mice. Regarding ADAM10, it tended to increase in TgAPP mice with rutin treatment. As for caspase-3 expression, TgAPP displayed an increase which was the opposite with rutin. Finally, the increase in expression of the inflammatory markers IL-1β and IFN-γ in TgAPP mice was lowered by both quercetin and rutin. Collectively, these findings suggest that, of the two flavonoids, rutin may be included in a day-to-day diet as a form of adjuvant therapy in AD.

## 1. Introduction

Decline in cognitive function is a fundamental clinical neurodegeneration symptom strictly related to age [1]. The impact of nutrition on age-associated cognitive decline is an increasingly growing topic, as it is a vital factor that can easily be modified. Pathological changes in the brain observed during cognitive decline take place well before any clinical manifestation, which mostly occur in old age. This provides a lengthy period of time to establish prevention strategies concerning age-related cognitive decline and dementia, which is a major public health concern [2]. For many years, intensive research on compounds of natural origin, found in day-to-day diets, has been carried out on cognitive-enhancing therapy [3].

One of the most common age-related neurodegenerative diseases is Alzheimer’s disease (AD), which is characterized by two neuropathological hallmarks: amyloid-β (Aβ) plaques and neurofibrillary tangles. In terms of research on animals, animal models can simulate the asymptomatic phase of AD by modifying the Aβ precursor protein (APP), for example [4,5].

It is remarkable that several bioactive phytochemicals derived from plants associated with various health benefits and decreased risk of many diseases have been screened for forming noncovalent complexes with the amyloid-β (Aβ) peptide [3].

Although numerous traditional medicines and natural dietary products have shown great progress toward AD pathology mitigation, we are fully aware of the limitations of AD animal models, since promising effects of those substances are not always replicable in human studies [6]. However, due to the difficulty of analyzing brain tissue in humans, especially at very early stages of progression, studies in rodent models are necessary. As a result, these experimental models may support the development of useful agents from traditional medicines and safe natural compounds to delay the progression of neurodegenerative diseases. Thus, testing natural compounds, found in day-to-day diets, for disease prevention and protection against the risk of AD, should be a priority.

Among these important dietary natural agents is quercetin, which is the main polyphenolic flavonoid in several fruits and vegetables [7]. Quercetin is mainly present in its glycoside form, i.e., rutin. For its part, rutin (quercetin-3-O-rutinoside) has shown profound effects on the various cellular functions that underpin several pathological conditions, namely antimicrobial, anticarcinogenic, antithrombotic, cardioprotective, and neuroprotective. These pharmacological effects are mainly associated with rutin’s anti-inflammatory and antioxidant activities. Due to its ability to cross the blood–brain barrier, and/or its metabolites, it has been demonstrated that rutin is able to alter both cognitive and behavioral symptoms of neurodegenerative diseases [8].

Our previous studies have found that both flavonoids, quercetin and rutin, affect various signaling pathways and molecular networks associated with the modulation of proteasome functions in neuroblastoma cells [9]. In addition, it has been demonstrated that BACE1 expression and APP processing are regulated by the ubiquitin–proteasome pathway [10] and that supplementation with reduced glutathione (GSH) protected neurons from proteasome inhibition [11]. GSH depletion in the brain is a common finding in patients with neurodegenerative diseases, such as AD, and can cause neurodegeneration prior to disease onset [12]. Ubiquitination of BACE1 and blocking the ubiquitin–proteasome pathway inhibits BACE1 degradation and, consequently, leads to increased production of BACE1 enzymatic activity [10]. 

Based on these findings, and that dietary habits and supplementation can affect the cellular redox status, we aimed to explore the effects of a diet containing quercetin or rutin on intracellular redox homeostasis of the brain (GSH/GSSG), its correlation with BACE1 activity, and APP expression in mice models of AD (bearing human Swedish mutation Amyloid Precursor Protein APP transgene, APPswe). 

Finally, as there is convincing evidence of an effect of flavonoid supplementations in improving specific cognitive domains and/or MRI findings [13], we attempted to mimic, in animals, an intervention by delivering a healthy diet containing moderate amounts of a particular potential active ingredient (quercetin or rutin) as an effective strategy for preventing the expression of AD markers.

## 2. Results

### 2.1. Genotyping of Mice

The TgAPP mouse colony was developed in our laboratory from Tg2576 heterozygous males and wild-type females. Genotyping of mice was performed to detect transgenic individuals. Of all the mice tested, approximately 40% were found to be transgenic (TgAPP).

The PCR products obtained were separated by electrophoresis in 1.5% agarose gels in 0.5X TBE (Tris-Borate-EDTA) buffer at 70 V (constant voltage) and then imaged by staining with GelRed (Millipore). The amplification profile for both transgenic and WT mice is shown in Figure 1.

### 2.2. Glutathione

In order to evaluate intracellular redox homeostasis in neurons in TgAPP mice, GSH and GSSG levels were quantified, and the GSH/GSSG ratio was determined as a marker of cellular-reducing power in both males and females (Figure 2). 

In the assessment of the effect of the transgene on the glutathione system, a decline in the cellular-reducing power (GSH/GSSG) was observed in the TgAPP mice with respect to WT animals, in both males and females, and in both areas of the brain (Figure 2C3,H3), especially in hippocampus. In both WT and TgAPP mice, the GSH/GSSG ratio was significantly lower in males than in females (Figure 2C3,H3; *p* < 0.05). While in TgAPP females this decline is the result of lower GSH levels (Figure 2C1,H1), in TgAPP males it is mostly attributed to an increase in GSSG levels (Figure 2C2,H2). 

Changes in GSH and GSSG levels of TgAPP mice, respectively, upon quercetin or rutin treatment, are more prominent in males than in females. It seems that quercetin tends to augment GSH levels (Figure 2C1,H1; *p* < 0.05) and rutin to lower GSSG levels (Figure 2C2,H2; *p* < 0.05).

Quercetin and rutin treatments, in both males and females, were able to reverse the fall in the ratio GSH/GSSG in hippocampus (Figure 2H3) where the recovery of redox power was significant versus the untreated TgAPP mice. In males, this index achieved similar values to those of WT mice in hippocampus (H3). In females, although this ratio is not raised up to that of the WT mice, treatment with quercetin and rutin enhanced it significantly in comparison to that of the TgAPP mice in their hippocampi (H3: quercetin, *p* < 0.001; rutin, *p* < 0.05). 

### 2.3. Thiobarbituric Acid Reactive Substances (TBARs)

Levels of TBARs were determined as a marker of lipid peroxidation. Using calibration curves, the results were expressed as malondialdehyde (MDA) concentration (Figure 3). 

Following APP overexpression, a significant increase in MDA levels when compared to WT mice was observed in both the cortex (Figure 3C) and the hippocampus (Figure 3H), and in both males and females (*p* < 0.001). 

In TgAPP females, both quercetin and rutin treatments almost restored MDA levels to the same as those of WT mice (Figure 3C,H). In TgAPP males, likewise, both quercetin and rutin treatments reinstate MDA levels to the same as those of WT mice in the cortex (Figure 3C), and are decreased even further in the hippocampus (Figure 3H). In both WT and TgAPP mice, untreated and flavonoid diet-treated, MDA levels were sex-dependent (Figure 3C,H; *p* < 0.05), except for the quercetin-treated TgAPP mice in hippocampus.

### 2.4. Enzyme Activity and Expression of Antioxidant Enzymes

To address whether regulation of the enzymatic activity or the gene expression of the main antioxidant enzymes, or both, occurs upon a quercetin or rutin diet, determination of the enzymatic activity and mRNA levels was performed. 

Figure 4 shows the enzyme activities of SOD, CAT, GR, and GPx, determined in female and male mice, in the cerebral cortex (Figure 4a) and the hippocampus (Figure 4b).

As a consequence of APP overexpression, only a significant decrease in CAT activity was observed in TgAPP mice compared to WT mice in both the cortex (Figure 4a(C2); *p* < 0.05) and the hippocampus (Figure 4b(H2); *p* < 0.05).

Quercetin treatment did not produce any significant variation in enzyme activities in comparison to TgAPP mice, in males or females, in the brain areas studied. In contrast, animals treated with rutin experienced an increase in CAT activity in the cortex (Figure 4a(C2)) and in GR activity in the hippocampus (Figure 4b(H3)) in both males and females (*p* < 0.05). Moreover, rutin increased hippocampal CAT activity in TgAPP males (Figure 4b(H2)) and GPx activity in females (Figure 4b(H4)). 

Figure 5 shows the gene expression of the main antioxidant enzymes, SOD, CAT, GR, and GPx, determined in female and male mice, in the cerebral cortex (Figure 5a) and the hippocampus (Figure 5b).

No differences in gene expression between TgAPP and WT mice are observed for the main antioxidant enzymes (Figure 5a,b). TgAPP males treated with rutin showed a significant increase in the expression of CAT in the hippocampus (Figure 5b(H2)). As for the hippocampal GPx, a similar pattern to CAT was observed, although the increase was not significant (Figure 5b(H4)). 

### 2.5. APP Processing: BACE1 and ADAM10

The results of BACE1 enzyme activity in both the cerebral cortex and the hippocampus in males and females are shown in Figure 6, expressed as percentages of activity with respect to untreated TgAPP mice.

In Figure 6, BACE1 enzyme activity in TgAPP mice was found to increase by around 10% when compared to WT mice, in both the brain areas under investigation and in both sexes, and was found to be statistically significant (*p* < 0.05).

The increase in activity observed in the transgenic mice was lowered by both quercetin and rutin treatments, both in the cortex and in the hippocampus. Nevertheless, it could still be noted that the rutin effect was slightly greater than that of quercetin in males.

Once the activity of BACE1 was known, we decided to carry out the gene expression study of APP, the main characteristic of the transgenic animal model, and its main processing enzymes: BACE1 and ADAM10. Figure 7 shows the results obtained in female and male mice, both in the cortex and the hippocampus.

In both sexes, a significant increase in APP expression greater than 85% was observed with respect to WT mice, demonstrating the overexpression of the gene both in the cerebral cortex (Figure 7C1; *p* < 0.05) and in the hippocampus (Figure 7H1; *p* < 0.05). Treatments with quercetin and rutin were able to reduce this expression by more than 45% (*p* < 0.05) for both male and female mice in both brain areas under investigation (Figure 7C1,H1), with the effects being more prominent in the hippocampus (Figure 7H1).

As for the BACE1 protein expression, though BACE1 activity was altered, there were no significant differences between transgenic and non-transgenic mice regardless of sex and flavonoid treatment examined.

Thus, we evaluated the ADAM10 expression involved in the non-amyloidogenic processing of APP. Although the changes in ADAM10 expression in TgAPP mice in comparison to WT mice were not statistically significant, a slight decrease was observed. Regarding the flavonoid treatments, rutin displayed an increasing trend in ADAM10 expression, both in males and females and in both areas of the brain (Figure 7C3,H3).

### 2.6. Expression of Caspase-3 and Caspase-6

TgAPP mice showed an increase in caspase-3 gene expression (Figure 8C1,H1), which was significant and greater than 30% compared to the hippocampi of WT mice (Figure 8H1; *p* < 0.05). As for caspase-6 expression, no differences were observed between transgenic and non-transgenic mice (Figure 8C2,H2). 

Quercetin and rutin treatments were able to lower caspase-3 mRNA levels in the hippocampus in a statistically significant manner (Figure 8H1; *p* < 0.05), with inhibition percentages of around 17% and 27% for female and male mice, respectively. In the cerebral cortex, significant differences were only observed in the treatment with rutin in males (Figure 8C1; *p* < 0.05). 

With regard to caspase-6, quercetin and rutin treatments did not exert any statistically significant effect in the cortex or in the hippocampus (Figure 8C2,H2), though in the latter the Caspase-6 in TgAPP males showed a tendency to decrease (Figure 8H2). 

### 2.7. Inflammation Markers

The results obtained for gene expression of the inflammatory mediators IL-1β, TNF-α, and IFN-γ are shown in Figure 9.

In the TgAPP, there was a significant increase in IL-1β gene expression of around 20% in the cortex and hippocampus in both sexes compared to WT mice (Figure 9C1,H1; *p* < 0.05). As regards TNF-α, although higher mRNA levels are shown in TgAPP, they are not statistically significant in relation to WT mice (Figure 9C2,H2). Regarding IFN-γ, there was an increase of around 30% in its expression in males, which was only statistically significant in the cortex (Figure 9C3; *p* < 0.05).

Treatments with quercetin and rutin, both in females and males, were able to diminish IL-1β expression in the cerebral cortex and hippocampus in comparison to control TgAPP mice (Figure 9C1,H1; *p* < 0.05), obtaining similar values to those of WT mice, and particularly lower in the hippocampi of male mice (Figure 9H1; *p* < 0.05). 

The overall effect of both flavonoid treatments on IL-1β expression was not observed with TNF-α nor with INF-γ. Thus, TgAPP males, upon quercetin treatment, underwent a significant decrease in hippocampal TNF-α (Figure 9H2; *p* < 0.05) and in cortical IFN-γ expression (Figure 9C3; *p* < 0.05).

### 2.8. Assessment of Degenerating Neurons and Its Projections

No characteristic signs of neurodegeneration were observed at the age at which the transgenic TgAPP mice were tested, compared to WT mice, nor did treatments with quercetin and rutin show any change for 4 weeks in comparison to TgAPP (Appendix A). 

### 2.9. Expression of Ionotropic Glutamate Receptors

No significant differences were found in receptor expression, comparing the values obtained for the control TgAPP mice with those obtained for the WT mice. There were also no notable effects on the expression of these ionotropic receptors in the presence of quercetin or rutin treatment (Appendix A).

## 3. Discussion

The purpose in our present study was to assess the impact of two flavonoids, quercetin and rutin, at the first stages of AD pathogenesis, regardless of their effect on neurodegeneration and/or cognitive function. The cortex and hippocampus were the areas of the brain under analysis, as they are the most affected brain structures in AD. It should be taken into account that quercetin and rutin were administered through a formulated diet containing either one of the two flavonoids, with the aim to mimic, in an AD animal model, the intake of a healthy human diet, containing an active ingredient. 

In particular, the transgene APPswe in the C57B6 mouse exerted a significant impact on GSH/GSSG ratio, MDA levels, antioxidant enzyme capacity, APP expression, BACE1 activity, and caspase-3 and IL-1β expression. Whilst APP mutations in humans generally result in typical AD, they are predominantly linked to solely amyloid pathology in APP transgenic mice and there is no noticeable neurodegeneration [14,15], as there were no characteristic signs observed in our transgenic mice TgAPP, contrary to the WT mice (Appendix A). Counterstaining with 4′-6-diamidino-2-phenylindole (DAPI) of hippocampal neurons allowed us to observe the nuclear morphology, as this compound is a fluorescent dye for nucleic acids. We did not observe fragmented or lobular nuclei, typically apoptotic; nor did we observe any remarkable differences comparing the hippocampal histological sections of the control transgenic line TgAPP with respect to the WT sections; nor did we observe any differences between the quercetin and rutin treatments with respect to the control TgAPP mice.

In the panel of AD biochemical features to be analyzed, we focused primarily on determining the GSH/GSSG ratio upon either one of the two flavonoid diets, since depletion of GSH levels represents one of the most important early biochemical markers in AD [16,17] and has been observed during its pathogenesis and disease progression. Measurement of brain GSH levels [18] and, more recently, blood GSH levels [19] have been promising as diagnostic markers for early stages of AD. Moreover, efforts have also been made to supplement endogenous GSH stores by themselves or their precursors [20,21,22]. In our study, a decline in the cellular reducing power (GSH/GSSG) was observed in the TgAPP mice with respect to WT animals, in both males and females, and in both areas of the brain. In cortex and hippocampus of both WT and TgAPP mice, the GSH/GSSG ratio was lower in male than in female. Quercetin and rutin diets significantly increased the GSH/GSSG ratio in comparison to untreated TgAPP mice, and this increase was more pronounced in the hippocampus. The changes in GSH and GSSG levels and GSH/GSSG ratio upon quercetin or rutin treatment of males, regarding increasing redox power, were more prominent than in females. The results from our determinations may reveal an important basis underlying sex-associated differences in Tg2576 mice in the susceptibility to the oxidative damage of macromolecules on one hand, since the glutathione system is a versatile reductant in multiple biological functions, and in the impact of preventive flavonoid diets in restoring its physiological status on the other hand. As we will see throughout this discussion, we have set the increase in the GSH/GSSG ratio as the main axis that might explain the set of effects observed in the TgAPP mice.

It has recently been proposed that the GSH/GSSG ratio, rather than simply functioning as a redox buffer, would instead operate as a main regulatory mechanism, allowing proteins to attain their native conformation and functionality by tightly controlling the thiol-disulphide balance of the cellular proteome. In short, the glutathione system arises as essential to preserve a healthy proteome, showing that disruption of glutathione redox homeostasis (i.e., genetically or pharmacologically) increases protein aggregation due to disturbances in the efficacy of autophagy [23]. Therefore, strategies aimed at maintaining glutathione redox homeostasis may have a therapeutic potential in diseases associated with protein aggregation, such as AD. Closely related to the preservation of the proteome is the ubiquitin–proteasome degradation machinery, which is involved in the pathogenesis of AD. The proteasome selectively degrades multiple substrates that are crucial in maintaining neuronal homeostasis, including the catabolism of oxidized and aggregated proteins. BACE1 undergoes ubiquitination, and it has been demonstrated that blocking the ubiquitin–proteasome pathway will inhibit BACE1 degradation and consequently lead to increased production of BACE1 enzymatic activity, more β-cleavage product C99, and increases in both Aβ_1-40_ and Aβ_1-42_ in neuronal and non-neuronal cells [10]. Our previous studies have found that both flavonoids, quercetin and rutin, affect various signaling pathways and molecular networks associated with modulation of proteasome function in neuroblastoma cells [9]. In addition, it has been demonstrated that neurons supplemented with reduced glutathione (GSH) recovered the proteasome activity and reduced aggregate formation [11], since the proteasome function is redox status-regulated [24]. Therefore, the increase in the GSH/GSSG ratio experienced by the animals upon having a quercetin or rutin diet is consistent with the modulation of proteasome by quercetin and rutin, demonstrated ex vivo previously. 

As previously mentioned, redox imbalance leads to highly oxidatively-modified proteins that tend to accumulate and create aggregates resulting in proteasome impairment [25]. Thus, given the crucial role of oxidative stress in the pathogenesis of AD, biomarkers of oxidative stress, including lipid peroxidation (MDA levels) and antioxidant enzymes, were assessed in the cortex and hippocampus in the TgAPP and WT mice. SOD, CAT, GR, and GPx are the most important antioxidant enzymes that act against oxygen free radicals and regulate the metabolism of free radicals in the body and play a role in the free radical scavenging system, protecting the cells in the body from lipid peroxidation. In our study, as a consequence of APP overexpression, a generalized decrease in antioxidant enzyme activities was observed in TgAPP mice compared to WT mice, being statistically significant for CAT. Consistent with reduced GSH levels, lipid peroxidation was significantly increased in the TgAPP mice. While the source of oxidative stress in human AD is highly complex and multifactorial, the amyloid pathology developed in mice seems to be sufficient to initiate the pathological process leading to increased oxidative stress in the brain [26]. Animals treated with rutin experienced an increase in CAT activity in the cortex and in GR activity in the hippocampus, in both males and females. Only animals treated with rutin experienced changes in gene expression of CAT and GR in the cortex and the hippocampus in both males and females, and GPx in the hippocampi of female mice. In this context, several natural compounds have been shown to affect the crosstalk between the proteasome and redox regulation. More precisely, quercetin is a known Nrf2 activator [27] which exhibits antioxidant properties through the stimulation of proteasome function, promoting increased oxidative stress resistance and conferring enhanced cell longevity [28]. 

Tissue-specific expression of BACE1 is critical for normal APP processing, and its dysregulation expression may play a role in AD pathogenesis. BACE1 is predominantly expressed in hippocampal neurons, the cortex, and the cerebellar granular layer [10]. It should be noted that earlier studies have shown that Swedish mutant APP transgenic mice had significantly increased brain levels of Aβ at a steady state [29], suggesting that BACE1 plays an essential role in the amyloidogenic pathway in AD pathogenesis and is a good therapeutic target for AD treatment. In our study, we observed a significant reduction of BACE1 activity upon quercetin and rutin treatments, which might contribute to the decrease of Aβ deposition in mice. We argue that more than solely operating as BACE1 inhibitors of the enzyme, quercetin and rutin might exert a reduction in BACE1 activity related to an increase in the ratio GSH/GSSG, based on the hypothesis of an enhancing recovery of proteasome activity. In this sense, it is known that targeting of BACE1 inhibitors to the β-cleavage site of APPswe (Swedish mutation) occurs before it reaches the plasma membrane, whereas APPwt (Wild-type) is processed in an early endosome originating at the cell surface. Therefore, BACE1 that cleaves APPwt is sometimes bound to the BACE1 inhibitor on the cell surface prior to APP processing, however, the enzyme that processes APPswe is not [30]. It is for this reason that the aberrant localization of APPswe processing might significantly lower the potency of quercetin and rutin as BACE1 inhibitors. Thus, we are more inclined to support that the BACE activity’s decreasing in this in vivo model is not so much due to the inhibition of the enzyme but to the increase in the GSH/GSSG ratio. In any case, reduced BACE1 activity could be interpreted as a putative attempt to reduce β-amyloid production in the TgAPP mice. As for the most remarkable effect of quercetin and rutin in the hippocampus on BACE1 activity attenuation, it is worth noting that the cortex has a significantly higher neuron density than the hippocampus [31], and a selective impairment of the proteasome in AD pathological phenotype makes the cortex more vulnerable and affected than the hippocampus [32].

After determining the effect of the treatments on BACE1 enzyme activity, we were interested in evaluating its expression. Curiously, no significant differences in BACE1 expression were found between TgAPP mice compared to WT mice and no noticeable changes were observed with quercetin or rutin treatment. Therefore, it seems that the increase in BACE1 enzyme activity is not associated with an increase in expression. In this context, it is remarkable that Apelt et al. [14] found an increase in cortical BACE1 activity in Tg2576 mice between ages of 9 and 13 months while the expression level of BACE1 protein and mRNA did not change with age. Furthermore, evidence has been found supporting that fibrillar amyloid Aβ_1–42_., rather than soluble amyloid Aβ_1–42_, is able to upregulate BACE1 protein expression, and thus small modifications in the ratio of amyloid isoforms may modulate amyloid aggregate conformations and cell damage [33]. Thus, the absence of change in BACE1 expression upon an increase of its activity that we found may account for the prevalence of soluble amyloid Aβ_1–42_ over the fibrillar amyloid Aβ_1–42_ isoform in our mouse model TgAPP.

Following the determination of gene expression of the enzymes involved in APP processing, we evaluated the effect of quercetin and rutin on the enzyme α-secretase involved in the non-amyloidogenic processing of APP. We focused on ADAM10 because it is the physiologically most important constitutive isoform of α-secretase. ADAM10 counteracts the generation of neurotoxic oligomeric Aβ plaques via cleaving APP within the Aβ domain to produce sAPPα and C-terminal fragment (α-CTF) [34,35]. Although the changes in ADAM10 expression found in our study were not statistically significant, a slight decrease in ADAM10 expression was observed in TgAPP mice relative to WT mice. Predominantly, rutin treatment showed a tendency to increase ADAM10 gene expression in both brain areas under study. Postina et al. [36] showed that the up-regulation of wild-type ADAM10 in the hippocampus of an AD mouse model mediated sAPPα secretion, leading to inhibition of Aβ plaques generation. The effect of quercetin has been studied in an aluminum chloride-induced AD rat model showing a significant enhancement of the α-secretase (ADAM10 and ADAM17) in the hippocampus compared to untreated ones. This indicates that quercetin possesses the potential to increase the non-amyloidogenic pathway through the activation of α-secretase genes [37]. Preclinical data reinforce the hypothesis that enhancing brain sAPPα levels is a potential strategy to improve AD-related symptoms and attenuate synaptic deficits. ADAM10 and BACE1 compete for the APPβ cleavage, therefore potentiating ADAM10 activity might inhibit the neurotoxic amyloid generation. Moreover, sAPPα can prevent the activation of the stress JNK-signaling pathway, leading to activation of NF-κB-induced phosphorylation activity, which leads to proteasome degradation [38]. Therefore, the formation and the accumulation of disease-related protein aggregates are significantly reduced, and the cellular proteasome activity is enhanced, thereby providing evidence for a function of sAPPα in the regulation of proteostasis [39]. Furthermore, it has been demonstrated that sAPPα specifically upregulates glutamate AMPA receptor synthesis and its trafficking [40]. In our study, we explored whether the slight increase of ADAM10 expression upon rutin treatment exerts some influence in glutamatergic synaptic transmission. As shown in the Appendix A, no significant effects on the expression of these ionotropic receptor were observed upon quercetin and rutin diets, perhaps due to a weak increase in ADAM10 expression, which is not sufficient for the upregulation of the AMPA receptor (Appendix A).

It should be taken into consideration that in vitro studies have shown a wide variety of ADAM10 substrates [41], and therefore, undesirable effects obtained by non-specific ADAM10-targeting might be found in cancer proliferation, cell adhesion, promotion of T cell/NK-cell precursor and inflammation, etc. [42]. To circumvent this constraint, our study suggests a strategy aimed at promoting the release of sAPPα in a more physiological manner. This approach might be based on a long-term intake of an active ingredient (quercetin or rutin), which is consumed through a healthy human diet. However, further studies are needed to find out whether the increase in ADAM10 is flavonoid dose-dependent and whether the potential beneficial effects outweigh putative side effects.

As for the expression of APPswe, although the insertion of the human APP transgene in the mouse genome guarantees that APPswe is overexpressed from birth, it has been reported that APP mRNA and protein hippocampal levels show significant fluctuations during the animal development, being maximal when mice are asymptomatic (1-month-old) and decreasing when full symptomatology occurs [43]. Notwithstanding this issue, APP expression both in the cortex and the hippocampus was significantly higher compared to that of WT mice in our study. Further treatment with quercetin or rutin was able to significantly reduce such expression for both male and female mice, in both areas of the brain. These findings are in line with those reported by Augustin et al. [44] who studied a standardised extract of *Ginkgo biloba* (Egb761), rich in flavonols such as quercetin, in 4-month-old female TgAPP mice, finding decreased APP mRNA and protein levels. Taking into consideration that upregulation of APP translational in Tg2576 mice occurs in the prodromal and early symptomatic stages [45], it is likely that a restoration of APP translation by quercetin or rutin might have taken place in our TgAPP mice and, likely, in an early symptomatic stage, resulting in reduction of cortical and hippocampal levels of APP, BACE1 activity, and caspase-3 activation. 

Furthermore, it has been reported elsewhere that in Tg2576 mice (in the absence of neuronal loss) there is an increase in caspase-3 activation in the hippocampus [46], as found in our study, at the onset of memory impairment, together with a reduction in dendritic spines prior to the deposition of extracellular amyloid [46]. There is evidence in support of non-apoptotic roles for caspases in the nervous system without neuronal death [47], and caspase-3 activity has been localized to dendritic spines where it may elevate calcineurin levels. In turn, the dephosphorylation of GluR1 subunit of AMPA-like receptors, triggered by calcineurin is thought to result in postsynaptic dysfunction. Our values of caspase-3 expression, as a consequence of transgenesis, are in agreement with those obtained by other researchers who reported an increase in caspase-3 expression at the level of dendritic spines in the hippocampus of TgAPP mice [48]. Since APP contains three distinct cleavage sites for caspase-3 in its amino acid sequence, two of which are located at the level of the extracellular domain and one in the intracellular C-terminal portion of the APP tail [49], hydrolysis of APP by caspase-3 may alter the proteolytic processing of APP in favor of the amyloidogenic pathway [50], leading to the release of a cytotoxic C-terminal-derived peptide of 31 amino acids in length (C31), for example [51]. This suggests that, since caspase-3 can mediate the amplification of toxic fragment release from APP, lowering caspase-3 expression by quercetin or rutin may allow for the clearance of aggregated protein. In addition, as mentioned earlier, we have explored the influence of both the increase and decrease of caspase-3 expression in glutamatergic synaptic transmission, based on the ability of calcineurin-activated caspase-3 to dephosphorylate the GluR1 subunit of AMPA receptors at the postsynaptic level. These molecular modifications alter glutamatergic synaptic transmission and neuronal plasticity at the level of dendritic spines in the hippocampus [48]. Theoretically, pharmacological inhibition of caspase-3 activity in TgAPP mice might save the AD-like phenotypes from a mechanism that drives synaptic failure. However, despite the augmentation in caspase-3 expression in our TgAPP mouse, we found no significant differences in AMPA receptor expression compared to that in WT mice, as mentioned earlier. It might be that the changes in caspase-3 expression are not prominent enough to produce significant modifications in AMPA receptor expression (Appendix A).

As for the values of caspase-6 expression, no significant differences were found between TgAPP and WT mice. Activation of caspase-6 has been identified as an important mediator of neuronal stress that cleaves important cytoskeletal proteins (Tau and α-tubulin), thus disrupting the ubiquitin–proteasome degradation of misfolded proteins, and a number of actin-regulating post-synaptic density proteins [52]. The unchanged expression of caspase-6 in our study agrees with the absence of characteristic signs of neurodegeneration at the age at which these transgenic mice were evaluated, compared with the WT mice (Appendix A).

A marked increase in neuroinflammatory mediators has been observed in AD patients, mainly around senile plaques [53,54,55]. Astrocytes are the main supplier of GSH to microglia and neurons. During chronic inflammation and oxidative stress, astrocytes release toxic inflammatory mediators and free radicals, accelerating activation of microglia and neurodegeneration [56]. It is worth noting that decreased intracellular glutathione is related to the activation of the inflammatory pathways, p38 MAP-kinase, Jun-N-terminal kinase (JNK), NF-κB, in human microglia and astrocytes [57]. In this regard, we decided to quantify the levels of IL-1β, IFN-γ, and TNF-α in our animal model and determine the effect of quercetin and rutin on them. As is known, inflammation promotes defective processing of Aβ peptide and APP, promoting Aβ peptide aggregation and in turn modifying Aβ reactivity [58]. Thus, in our study, we observed that TgAPP mice had increased mRNA levels of the pro-inflammatory mediators IL-1β, TNF-α and IFN-γ, compared to WT mice, showing that overexpression of APPswe might induce neuro-inflammatory cascades triggering a series of molecular pathways in glia and neurons, which would activate the inflammatory response. Quercetin and rutin were able to attenuate IL-1β gene expression in both males and females and in the brain areas studied. Several pieces of evidence support the anti-inflammatory effect exerted by quercetin at the CNS level, as it may inhibit the activation of transcription factors such as the nuclear factor-kappa B (NF-κB) [59], involved in the induction of iNOS, and therefore, decrease the release of mediators such as IL-1β, TNF-α and IFN-γ [60]. Regarding the impact of GSH on the inflammatory response, it should be noted that GSH is involved in the maintenance of optimal cytokine levels in such a way that the expression of pro-inflammatory cytokines (TNF-α, IL-1β, and IL-6) are increased due to GSH depletion, whereas the expression of anti-inflammatory cytokines (i.e., IL-10) remained unaltered. This GSH homeostasis alteration happens due to upregulation in NF-κB and JNK signaling pathway which could be the feasible apoptotic pathway towards neuronal cell death [61]. In our study, down-regulation of NF-κB by quercetin and rutin might be a plausible mechanism to recover the GSH/GSSG homeostasis and therefore the cause of the balance between pro-inflammatory and anti-inflammatory cytokines. Lastly, since BACE1 promotor has an NF-κB binding site, inflammation-induced activation of NF-κB facilitates the upregulation of BACE1 expression, and subsequently increases Aβ production [62]. Thus, if down-regulation of NF-κB occurs upon quercetin and rutin diets, BACE1 activity would decrease as a result of the release regulation of pro-inflammatory and not anti-inflammatory cytokines.

## 4. Material and Methods

### 4.1. Experimental Animals

A transgenic mouse (Tg2576, B6;SJL-Tg(APPswe)2576 Kha) that expresses the Swedish double mutation of human amyloid precursor protein (hAPP) was used as the animal model of experimental AD [14]. The mouse is a knock-in heterozygote line which expresses the human AβPP695 isoform with the double Swedish mutation (K670N/M671L; Lys670→Asn and Met671→Leu) under the control of the hamster prion protein promoter [63]. As a result, this mouse exhibits levels of human amyloid-β precursor protein (Aβ PP), six times greater than that of a mouse’s Aβ PP levels. In addition, this mouse shows higher levels of Aβ40 and Aβ42. Aβ deposits begin at 9 months of age [63]. Within the Tg2576 hippocampus and cortex, APPswe transgene expression is primarily neuronal [64].

As a negative control, wild-type (WT) mice from the same colony [65,66] were used. The Tg2576 (B6;SJL-Tg(APPswe)2576 Kha) mouse colony was developed in our laboratory from Tg2576 heterozygous males and wild-type females. The transgenic parents were donated by Dr. Diana Frechilla from the Neuroscience Division at the Centre for Applied Medical Research at the University of Navarra (Pamplona, Spain) [67].

Animals were housed in individual ventilated cages and kept at 22–24 °C on a 12-h light/dark cycle in 50–60% humidity. Animal protocols were approved by the Institutional Animal Care and Use Committee (IACUC) at the Complutense University of Madrid and were in full accordance with the European Directive 2010/63/on the protection of animals used for scientific purposes and Spanish legislation on Animal Welfare (Royal Decree 53/2013, 1 February 2013). 

### 4.2. Genotyping Analyses of Mice

Transgenicity was determined within 30 days of birth by tail biopsy. Genotyping analyses of animals were carried out by PCR. Considering that Tg2576 (TgAPP) is a heterozygous line, the insertion gene (PrP, from prion protein) was used as a positive reaction control. Genomic DNA was extracted from mouse tails digested with proteinase K (0.1 μg/μL) in NID buffer (50 mM KCl, 50 mM Tris-HCl pH 8.3, 50 mM MgCl_2_, 0.05% gelatin, 0.45% NP-40 and 0.4% Tween 20) at 56 °C for 3 h and shaken. DNA fragments were precipitated with isopropanol and washed with 70% ethanol. DNA precipitates were dissolved in 30 μL of TE buffer (10 mM Tris-1 mM EDTA). The purity and concentration of DNA was determined at 260 and 280 nm.

The PrP and APP genes were amplified by PCR. Sequences of primers used to screen the transgenic mice were as follows: PrP forward: CCTCTTTGTGACTATGTGGACTGATGTCGG; PrP reverse: GTGGATACCCCCTCCCCCAGCCTAGACC; APP reverse: CCAGATCTCTGAAGTGAAGATGGATG. The steps of the PCR reaction were as follows: denaturation at 94 °C for 90 s, 39 cycles at 60 °C for 60 s and 72 °C for 90 s, then final extension at 72 °C for 7 min.

In all cases, negative controls (without DNA mold) and positive controls of the APP gene were considered. The PCR products obtained were separated by electrophoresis in 1.5% agarose gels in 0.5X TBE (Tris-Borate-EDTA, Merck KGaA, Darmstadt, Germany) buffer at 70 V (constant voltage) and then imaged by staining with GelRed (Millipore). 

### 4.3. Animal Treatments

TgAPP mice and wild-type littermates, both aged 6–7 weeks with an initial body weight of 16.2 ± 0.8 g, were randomized into the following four groups (n = 8/group): (a) Untreated TgAPP; (b) Quercetin-treated TgAPP; (c) Rutin-treated TgAPP; and (d) Untreated wild-type. Since both male and female mice were studied, two sets of groups were established. At the age of 45 weeks, the mice started to be treated with quercetin or rutin for 4 weeks. 

Quercetin (3,3′,4′,5,7-pentahydroxyflavone) and rutin hydrate (quercetin-3-O-rutinoside hydrate) were ≥95% pure and purchased from Sigma–Aldrich. Each one of the flavonoids was incorporated into a standard diet (Harlan Ibérica, Barcelona, Spain) at a concentration of 200 ppm, corresponding to an intake of 30 mg flavonoid/kg body weight/day. The untreated mice received exclusively the un-supplemented standard diet. Diets and water were provided for *ad libitum* intake. 

### 4.4. Brain Tissue Preparation for Biochemical and Histological Assays

At the end of treatment, mice were fasted overnight, they were euthanized by means of cervical dislocation, and the entire brain was quickly removed. The brain was rinsed in saline at 4 °C and the arachnoid membrane was carefully removed. Then, the hippocampus and cortex were isolated. Samples were immediately stored at −80 °C until further use. 

The entire brains of some animals were used for obtaining histological sections, for which, once euthanized by means of cervical dislocation, brains were frozen by immersion in isopentane at −80 °C. Immediately afterwards, coronal sections of the brain (30 μm of thickness) were made from the olfactory bulb to the cerebellum, 120 μm apart in a cryostat (Leica CM1850, Nussloch, Germany). The whole procedure was performed at −20 °C. Histological sections were collected on slides and kept at −80 °C until analysis (See Appendix A).

### 4.5. Glutathione

For glutathione tests, cerebral cortex and hippocampus samples were homogenized in a redox-quenching buffer-5% Trichloride acetic acid (RQB-5% TCA) (previously bubbled with N_2_ for 15 min on ice) at a concentration of 25 mg/mL (*w*/*v*). Samples were resuspended by sonication for 10 s, then centrifuged at 12,000× *g* for 10 min at 4 °C, and supernatants were collected. 

Then, in the supernatant obtained, spectrofluorometric methods were adopted to determine GSH and GSSG levels using the o-phthalaldehyde method, described by Senft et al. [68]. GSH and GSSG values were corrected for spontaneous reaction in the absence of biological sample. In both cases, supernatants were incubated for 30 min at room temperature and afterwards fluorescence was measured using a FLUOSTAR microplate reader (BMG LABTECH, Ortenberg, Baden-Württemberg, Germany), with the excitation filter set at 360 nm (bandwidth 5 nm) and the emission filter set at 460 nm (bandwidth 5 nm). The concentration of GSH and GSSG in each sample was interpolated from known GSH standards. Concentrations of both GSH and GSSG were expressed as nmol GSH/mg protein, which allowed for the calculation of the glutathione redox ratio GSH/GSSG.

The remaining pellets were vortexed until completely dissolved in 240 μL of 0.1 M NaOH to measure protein concentration by the bicinchoninic acid (BCA) method, using bovine serum albumin as a standard.

### 4.6. Thiobarbituric Acid-Reactive Substances (TBARs) 

The content of TBARs was used as an index of lipoperoxidation. In brain tissue, 50 mM phosphate buffer (pH 7.4) was added to a concentration of 25 mg/mL (*w*/*v*) and the suspension was homogenized by sonication for 10 s. To 30 μL of the homogenate, 250 μL of 1% phosphoric acid and 75 μL of 0.6% thiobarbituric acid (TBA) were added. The reagent mixture was incubated at 100 °C in a water bath for 45 min, after which it was cooled in an ice bath and then centrifuged at 3000× *g* for 10 min at 4 °C. A volume of 150 μL of supernatant was taken from each sample. Fluorescence was measured using a FLUOSTAR microplate reader (BMG LABTECH, Ortenberg, Baden-Württemberg, Germany) with the excitation filter set at 485 nm (bandwidth 5 nm) and the emission filter set at 530 nm (bandwidth 5 nm). A calibration curve was prepared using malondialdehyde (MDA) as a standard. The results were expressed in pmol MDA/mg protein.

### 4.7. Enzymatic Activity of the Main Antioxidant Enzymes

For the determination of enzyme activity in brain tissue, a lysis buffer containing 50 mM phosphate buffer (pH 7.4) and antiproteases (1 mM EDTA, 1 mM PMSF, 1 g/mL pepstatin and 1 g/mL leupeptin) was added to a concentration of 50 mg/mL (*w*/*v*). Then, suspension was sonicated for 30 s in an ice bath, and the homogenate was centrifuged at 10,000× *g* for 15 min at 4 °C. Supernatants were collected for the determination of the enzymatic activity of the antioxidant enzymes.

Superoxide dismutase (SOD) activity was measured by following the inhibition of pyrogallol autoxidation at 420 nm [69]. One unit of enzyme was defined as the amount of enzyme required to inhibit the rate of pyrogallol autoxidation by 50%. The SOD enzymatic activity was expressed as international units (IU)/mg protein. Catalase (CAT) activity was measured in Triton-X-100 (1%, *v*/*v*)-treated supernatants by following hydrogen peroxide (H_2_O_2_) disappearance at 240 nm [70], and enzyme activity was reported as substrate (μmol H_2_O_2_) transformed/min **∙** mg protein. Total glutathione peroxidase (GPx) was determined following NADPH oxidation at 340 nm in the presence of excess GR, GSH, and cumene hydroperoxide [71]. GPx activity was expressed as substrate (nmol NADPH) transformed/min mg protein. Glutathione reductase (GR) activity was analyzed following NADPH oxidation at 340 nm in the presence of GSSG [72] and expressed as substrate (nmol NADPH) transformed/min **∙** mg protein. GR and both GPx activities were corrected for spontaneous reaction in the absence of biological samples (in the absence of enzyme).

### 4.8. BACE1 Activity Test

The ΒACE1 test protocol involves the use of a secretase-specific substrate (peptide) which is conjugated to two reporter molecules, namely EDANS and DABCYL, which results in the release of a fluorescent signal [73,74]. The BACE1 activity was measured both in the cortex and hippocampus lysates. The reaction was carried out at 37 °C for 1 h using 10 μM substrate in 50 mM sodium acetate buffer (pH 4.5). Fluorescence intensity measurements were done using a FLUOSTAR microplate reader (BMG LABTECH, Ortenberg, Baden-Württemberg, Germany) with the excitation filter set at 360 nm (bandwidth 5 nm) and the emission filter set at 530 nm (bandwidth 5 nm). The level of secretase enzymatic activity is proportional to the fluorometric reaction, and the data are expressed as x-fold increase in fluorescence over that of background controls (reactions in the absence of substrate or tissue). The BACE1 activity was normalized with protein concentration. The mice’s BACE1 activity, quercetin or rutin-treated, was expressed as the percentage of activity of that of TgAPP control mice. 

### 4.9. RT-PCR Gene Expression of the Main Antioxidant Enzymes, APP, BACE1, ADAM10, Caspase-3 and Caspase-6 and Inflammatory Cytokines

#### 4.9.1. Total RNA Extraction and Purification

We analyzed the different areas of the brain, namely the cortex and hippocampus, stored at −80 °C. To a known amount of brain tissue, Triomol^®^ lysis buffer was added at a ratio of 1:10 (*w*/*v*). Samples were homogenized for 30 s using a Cordless motor (Pellet pestle, Sigma-Aldrich), and incubated for 5 min at 25 °C to allow for complete dissociation of nucleoprotein complexes. Then, 0.2 mL of chloroform was added for each mL of Triomol^®^ lysis buffer used. The tubes were shaken vigorously for 15 s and incubated at 25 °C for 3 min. Then, they were centrifuged at 11,000× *g* for 15 min at 4 °C. After centrifugation, three phases were obtained, with RNA in the upper phase.

To isolate the RNA, the upper phase was transferred to another tube and precipitated by adding 0.5 mL isopropanol. After thorough mixing of isopropanol and aqueous solution by inversion, the mixture was incubated at room temperature for 10 min to promote precipitation, and centrifuged at 12,000× *g* for 10 min at 4 °C. The supernatants were removed, and the pellets were washed with 75% ethanol and centrifuged at 7500× *g* for 5 min at 4 °C. The pellets were dried at room temperature and dissolved in 50 μL of DEPC-treated water. To remove traces of DNA, 2.5 μL of DNase (RNase-free) was added and incubated at 37 °C for 30 min. Finally, samples were incubated at 64 °C for 5 min to inactivate the DNase.

Subsequently, the concentrations of RNA were measured in a UV-VIS spectrophotometer (BMG LABTECH, Ortenberg, Baden-Württemberg, Germany) at 260 nm and the purity was assessed considering the absorbance ratio at 260 and 280 nm (A260/A280).

The determination of RNA integrity and purity was performed by electrophoresis in a 1% agarose gel stained with GelRed and visualized under UV light, where, if the RNA was intact, two upper bands corresponding to ribosomal RNA (28S and 18S) and two lower bands corresponding to transfer RNA (tRNA) and 5S ribosomal RNA had to be observed.

#### 4.9.2. Complementary DNA (cDNA) Synthesis 

cDNA is much more stable than RNA and therefore allows for more convenient and safer sample handling. The cDNA was synthesized from mRNA by retrotranscription using the First Strand cDNA Synthesis Kit for RT-qPCR (Fermentas Life Sciences).

In order to carry out the retrotranscription for cDNA synthesis to 2 μg of RNA, 11 μL of DEPC-treated water and 1 μL of 10X Random primers were added. Then, the mixture was incubated at 65 °C for 10 min to denature the RNA. After this time, the tubes were immediately brought to 4 °C for 5 min to avoid renaturation of the RNA. The reagent mix for cDNA synthesis is shown in Appendix A.

Eight μL of the reaction mixture was added to each sample. The entire volume was brought to the bottom of the tubes and incubated at 42 °C for 60 min. Finally, the reaction was stopped by inactivating the reverse transcriptase by heating it at 70 °C for 10 min.

#### 4.9.3. Real-Time PCR

The main feature of real-time PCR is that the analysis of the products takes place during the amplification process by determining the fluorescence. In this way, the amplification and detection processes occur simultaneously in the same tube or vial without the need for any further action. For real-time PCR, thermal cyclers are used, which can amplify and detect fluorescence simultaneously. We utilized the LightCycler real-time thermal cycler (Roche Diagnostics, Mannheim, Germany).

Appendix A lists the reagents required for real-time PCR, using sequence-specific primers and DNA-binding dye (SYBR Green I, Roche Molecular Systems, Inc., Rotkreuz, Switzerland) as a detection system.

For the design of the primers for the different quantified markers, the Primer3Plus bioinformatics program (http://www.bioinformatics.nl/cgi-bin/primer3plus/primer3plus.cgi, accessed on 21 January 2023) was used, for which we took the cDNA sequences of the genes of interest from the Medline open-access database (http://www.ncbi.nlm.nih.gov/entrez, accessed on 21 January 2023). The primers were supplied by Merck (Sigma-Aldrich). The hybridization temperature and the sequence of the different primers used are shown in Appendix A.

The reaction conditions for the amplification of the genes of interest are shown in Appendix A.

Finally, the samples were subjected to a melting program: 95 °C for 15 s, 65 °C for 30 s, and up to 98 °C at a rate of 0.1 °C/s with continuous fluorescence recording.

For the quantification of cDNA levels, the cycle threshold (Ct) comparison method [75] was used, using GADPH as a housekeeper. The amplification of the housekeeper was done in parallel with the analyzed gene. Ct values were calculated using the 4.0 software provided by LightCycler (Roche Diagnostics, Mannheim, Germany). The software allows distinguishing between fluorescence due to sample amplification and due to background. Melting curves were also recorded. Determination of the melting temperature of the amplified fragment allowed for characterization of the amplified product. The size of the bands was checked on a 1.5% agarose gel.

The variation of the expression of the gene under study with the quercetin or rutin treatment was expressed as a function of the control TgAPP (mice without treatment) and normalizing this expression with the levels of GADPH. The *Change Fold* (2^−ΔΔCt^) represents the number of times that the gene of interest is modified under the particular treatment with respect to the control mice.

### 4.10. Statistical Analyses

All tests were performed at least in duplicate and in three different experiments. The results obtained are expressed as the mean ± standard error. One-way analysis of variance (ANOVA) was performed once the data were tested and demonstrated that it fits a normal distribution. The Newman–Keuls multiple comparison post-hoc test was run, examining mean differences between groups. Values of *p* < 0.05 were considered significant. SigmaPlot 11.0 software was used for statistical analyses.

## 5. Conclusions

Dietary habits and supplementation can affect the cellular redox status. On this basis, we aimed to ameliorate the cellular redox homeostasis in an AD mouse model by a flavonoid diet containing quercetin or rutin in order to alleviate amyloid pathology, considering the interplay between cellular redox status and proteasome-dependent amyloid features in asymptomatic AD. Our datasets are relevant, since the flavonoid effects displayed in the TgAPP mouse model are consistent with those reported earlier in our in vitro and ex vivo models.

In conclusion, our findings show that initiating a diet treatment at the asymptomatic stage or at the onset of AD-like symptoms might reinstate cellular redox status and APP physiological processing via concurrent regularization of APP expression and BACE1 activity. 

Although it is difficult to extrapolate our findings to the human condition, they may have broad implications for the human response to future therapeutics. Of the two flavonoids, rutin, with an overall more prominent in vivo effects, seems to be most suitable to be included in a day-to-day diet as an adjuvant therapy in AD, based on the augmentation on intracellular redox homeostasis of the brain.

## Figures and Tables

**Figure 1 ijms-24-04863-f001:**
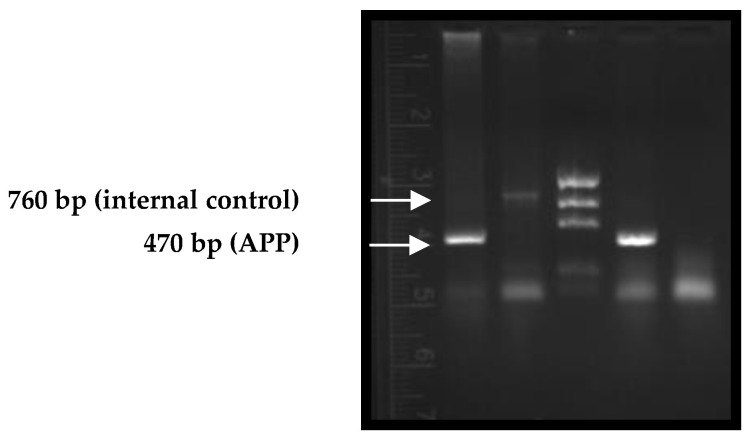
Genotyping of mice by PCR. (1) and (4) TgAPP mice (470 bp amplicon for the APP gen); (2) Wild-type (WT) mice (760 bp amplicon for the PrP gen); (3) MassRuler DNA ladder #170-8207; (5) Control reaction (without DNA mold).

**Figure 2 ijms-24-04863-f002:**
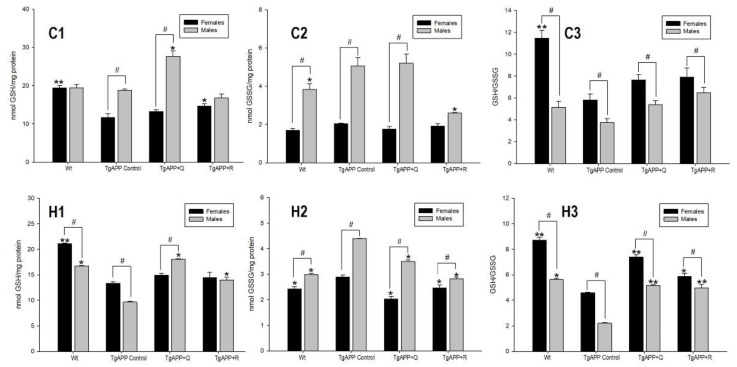
Effect of quercetin (Q) and rutin (R) on GSH (C1, H1), GSSG (C2, H2), and GSH/GSSG (C3, H3) levels. Cortex (C) and Hippocampus (H) (1) WT; (2) TgAPP control; (3) TgAPP + Q; (4) TgAPP + R. * *p* < 0.05 or ** *p* < 0.001 vs. TgAPP control (ANOVA followed by the post-hoc Newman–Keuls multiple comparison test); # *p* < 0.05 (Student *t*-test).

**Figure 3 ijms-24-04863-f003:**
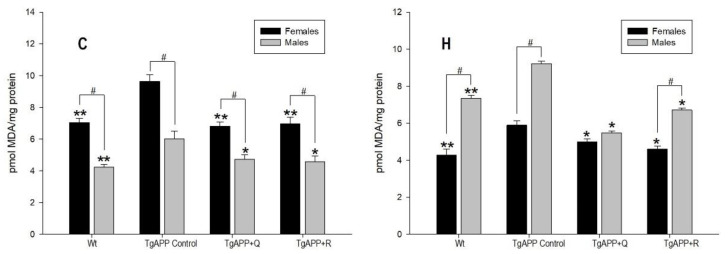
Effect of quercetin (Q) and rutin (R) on MDA levels. Cortex (C) and Hippocampus (H). (1) WT; (2) TgAPP control; (3) TgAPP + Q; (4) TgAPP + R. * *p* < 0.05 or ** *p* < 0.001 vs. TgAPP control (ANOVA followed by the post-hoc Newman–Keuls multiple comparison test); # *p* < 0.05 (Student *t*-test).

**Figure 4 ijms-24-04863-f004:**
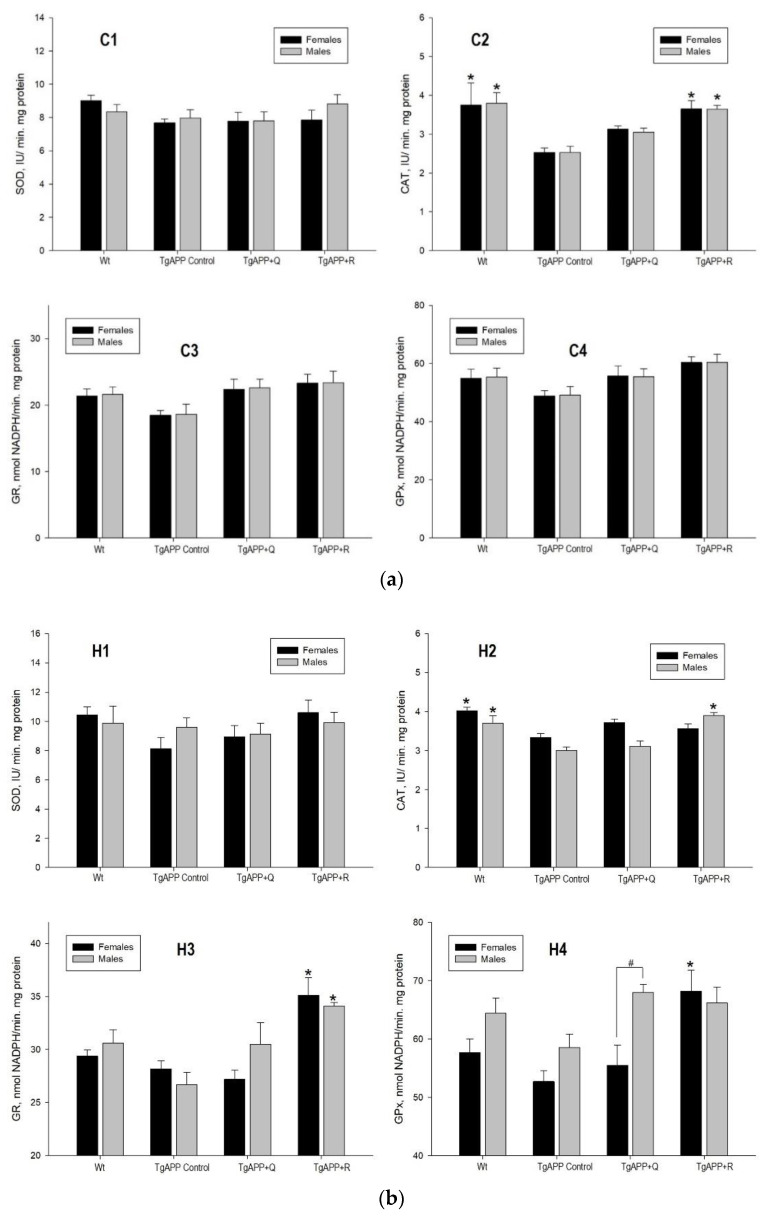
(**a**). Effect of quercetin (Q) and rutin (R) on the enzymatic activities of SOD (C1), CAT (C2), GR (C3), and GPx (C4) in Cortex (C). (1) WT; (2) TgAPP control; (3) TgAPP + Q; (4) TgAPP + R. * *p* < 0.05 vs. TgAPP control (ANOVA followed by the post-hoc Newman–Keuls multiple comparison test). (**b**). Effect of quercetin (Q) and rutin (R) on the enzymatic activities of SOD (H1), CAT (H2), GR (H3), and GPx (H4) in the Hippocampus (H). (1) WT; (2) TgAPP control; (3) TgAPP + Q; (4) TgAPP + R. * *p* < 0.05 vs. TgAPP control (ANOVA followed by the post-hoc Newman–Keuls multiple comparison test); # *p* < 0.05 (Student *t*-test).

**Figure 5 ijms-24-04863-f005:**
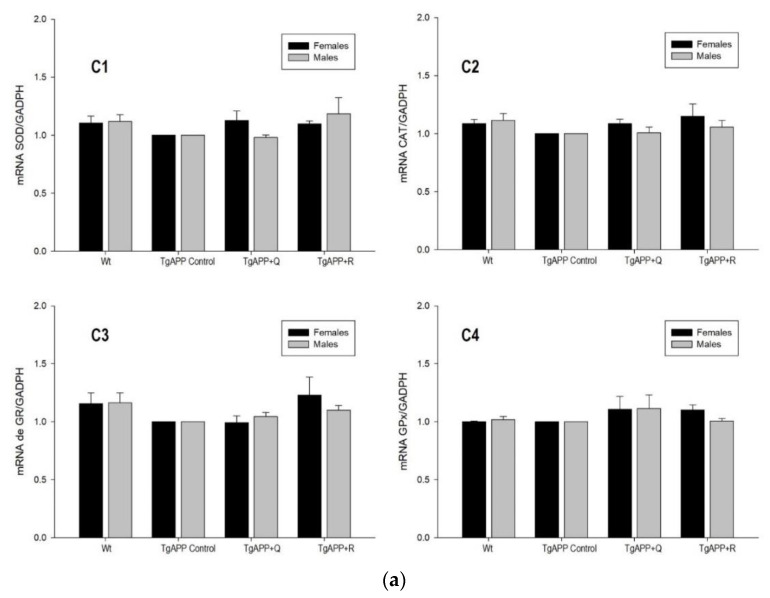
(**a**). Effect of quercetin (Q) and rutin (R) on SOD (C1), CAT (C2), GR (C3), and GPx (C4) expression in Cortex (C). (1) WT; (2) TgAPP control; (3) TgAPP + Q; (4) TgAPP + R. (**b**). Effect of quercetin (Q) and rutin (R) on SOD (H1), CAT (H2), GR (H3), and GPx (H4) expression in the Hippocampus (H). (1) WT; (2) TgAPP control; (3) TgAPP + Q; (4) TgAPP + R. * *p* < 0.05 vs. TgAPP control (ANOVA followed by the post-hoc Newman–Keuls multiple comparison test).

**Figure 6 ijms-24-04863-f006:**
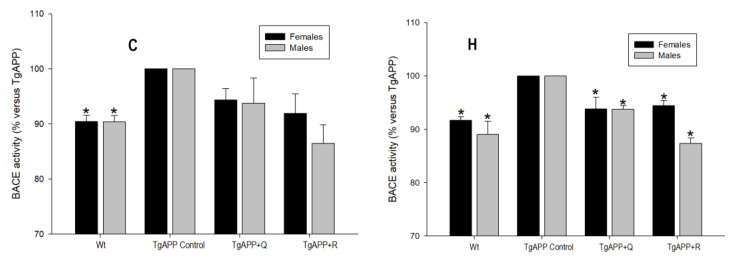
Effect of quercetin (Q) and rutin (R) on BACE1 activity in the Cortex (C) and the Hippocampus (H). (1) WT; (2) TgAPP control; (3) TgAPP + Q; (4) TgAPP + R. * *p* < 0.05 vs. TgAPP control (ANOVA followed by the post-hoc Newman–Keuls multiple comparison test).

**Figure 7 ijms-24-04863-f007:**
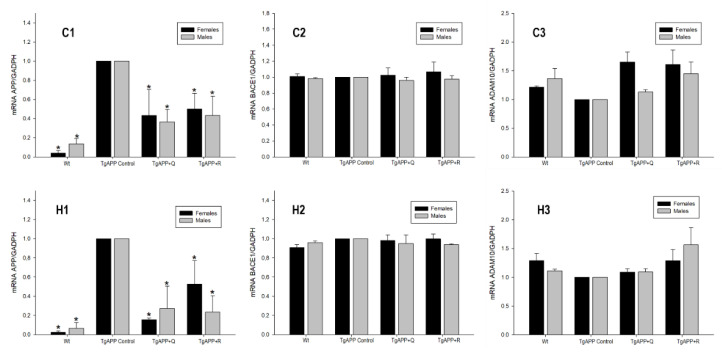
Effect of quercetin (Q) and rutin (R) on APP (C1, H1), BACE1 (C2, H2), and ADAM10 (C3, H3) expression in the Cortex (C) and the Hippocampus (H). (1) WT; (2) TgAPP control; (3) TgAPP + Q; (4) TgAPP + R. * *p* < 0.05 vs. TgAPP control (ANOVA followed by the post-hoc Newman–Keuls multiple comparison test).

**Figure 8 ijms-24-04863-f008:**
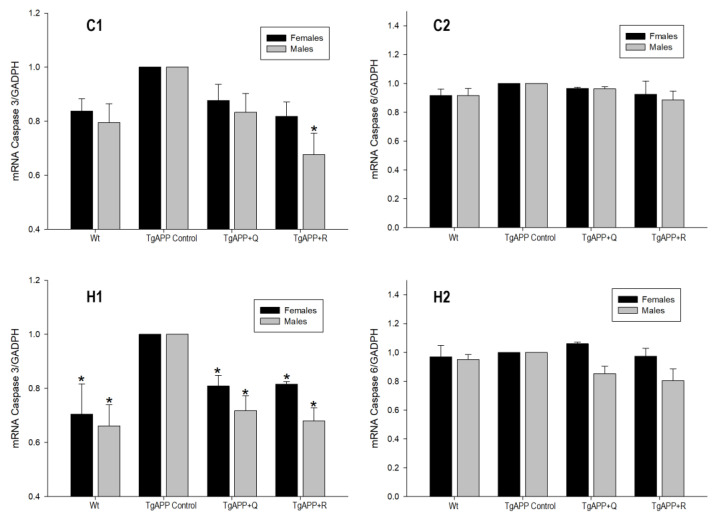
Effect of quercetin (Q) and rutin (R) on Caspase-3 (C1, H1) and Caspase-6 (C2, H2) expression in the Cortex (C) and the Hippocampus (H). (1) WT; (2) TgAPP control; (3) TgAPP + Q; (4) TgAPP + R. * *p* < 0.05 vs. TgAPP control (ANOVA followed by the post-hoc Newman–Keuls multiple comparison test).

**Figure 9 ijms-24-04863-f009:**
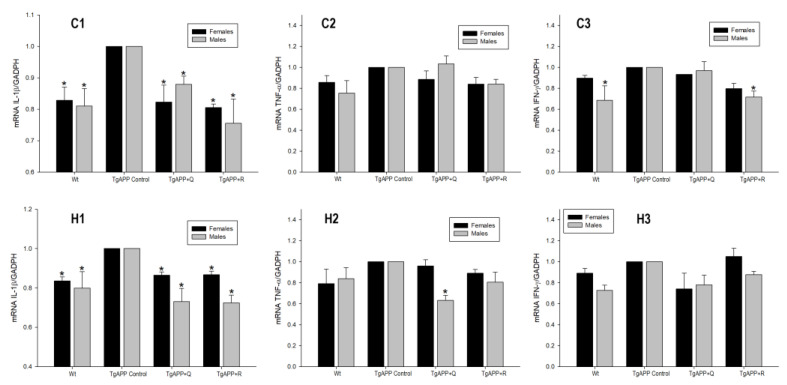
Effect of quercetin (Q) and rutin (R) on IL-1β (C1, H1), TNF-α (C2, H2), and IFN-γ (C3, H3) expression in the cortex (C) and the hippocampus (H). (1) WT; (2) TgAPP control; (3) TgAPP + Q; (4) TgAPP + R. * *p* < 0.05 vs. TgAPP control.

## Data Availability

The data that support the findings of this study are available from the corresponding author upon reasonable request.

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
