# Peer review of "A Diet Containing Rutin Ameliorates Brain Intracellular Redox Homeostasis in a Mouse Model of Alzheimer’s Disease"

_ijms, 2023, doi:10.3390/ijms24054863_

Round 1

Reviewer 1 Report (New Reviewer)

In this manuscript, the authors treated the AD mice with two flavonoids to investigate how these two compounds could modulate AD-related biomarkers early in the disease. They found flavonoids can ameliorate brain intracellular redox homeostasis in AD transgenic mice. Overall, the manuscript is clear and complete. But I think there are a few points that the authors can improve:

1. In Figure 2, the data shows a big difference between the male and female mice. The authors should explain more about it.

2. In most cases, the authors only used the mRNA level to characterize the protein expression, which I think is insufficient. Instead, the authors should use different antibodies to do immunofluorescence staining or Western to quantify the protein level. Lots of things may happen during transcription. 

Author Response

Dear reviewer,

Please see the attachment. Thank your very much.

Sagrario Martín-Aragon

Reviewer 2 Report (New Reviewer)

The article by Paloma Bermejo-Bescóset al, describes the effect of a diet containing rutin ameliorates brain intracellular redox  homeostasis in a mouse model of Alzheimer’s disease. 

The manuscript is original and of interest in its field. 

 I recommend that the paper be accepted with minor revision:

a) The authors should better check the manuscript for any typographical errors.

b) Please clarify how the authors incorporate quercetin into the diet

c) please remove the white page (10 of 25)

d) this referee would like to have a more in-depth comment on the differences in results between males and females

e) this referee would like to know from the authors whether there are differences between males and females and in which parameters

Author Response

Dear reviewer,

Please see the attachment. Thank you very much.

Sagrario Martin-Aragon

Reviewer 3 Report (Previous Reviewer 2)

The authors have correctly revised the article and it is ready for publication in IJMS

Author Response

Dear reviewer,

Please see the attachment. Thank you very much.

Sagrario Martin-Aragon

Round 2

Reviewer 1 Report (New Reviewer)

The authors have fully addressed my concerns. The manuscript can be accepted by IJMS now. 

This manuscript is a resubmission of an earlier submission. The following is a list of the peer review reports and author responses from that submission.

Round 1

Reviewer 1 Report

This research article investigated how a diet containing flavonoids, quercetin or rutin, influences the level of various enzymes and bioactive compounds contributing to intracellular redox homeostasis in the brain. Implementing different dietary products for making up a specific diet was a trend within society for many years, and research efforts were made to figure out a basis for some effects and build up a tendency to a diet rationale. In this paper, the authors used molecular biology approaches and biochemical assays to find the difference in various protein/enzyme levels in the brain, which apparently contribute to ageing and can also occur in Alzheimer’s disease. The authors used a transgenic TgAPP mouse line (bearing human Swedish mutation APP transgene, APPswe) to study the effect of flavonoids taken as a supplement for a prolonged time. Based on the obtained data, the authors made a conclusion about the beneficial effects of flavonoids, especially rutin, in anti-Alzheimer’s disease treatment. Although a large number of experimental data sets are present, the experimental design has severe flaws that raise criticism regarding the data interpretation, hence conclusions. Among the main issues, there are follows.

  1. The authors’ choice of animal model raises significant concern about its relevance for ageing and neurodegeneration, in principle. Basically, there are numerous phenotypes of Alzheimer’s disease, and either of those has very characteristic neuropathology manifested in systemic dysfunctions and impaired neural cell morphology at the tissue level. One can expect that the authors show first a clear phenotype of neuropathology, which they try next to alleviate. However, the authors’ data on the absence of any characteristic signs of neurodegeneration in mutant mice (Suppl. Figure 10) and their attempts to discuss the fact of no clear pathology seen (lines 492-504) also support this uncertainty. How relevant are then the effects of flavonoid treatment on Alzheimer’s disease? What pathology could the data be relevant for?
  2. Further to the above, the authors listed the pros and cons of using animal models for studies of Alzheimer’s disease in the Introduction (lines 74-81) to argue for an impossible use of the human brains for mechanistic studies. However, human stem cell models have been commonly employed in the field – they perfectly provide a tool for human-relevant testing of new treatments and drug screening. Given the uncertainty of the used animal model, it appears that human iPSC cultures (or spheroids) would be a much more appropriate and beneficial model for testing the effects of flavonoids against Alzheimer’s disease. The cells can be treated with exogenous beta-amyloid to model the neuropathology at any stage of the disease (early-onset or advanced pathology).
  3. It remains unclear the route of the treatment. Was it by supplementing quercetin or rutin to food or drinking water? How did the authors manage strict control of drug intake to ensure the concentration they reported in the methods (lines 149-159)?
  4. Further to managing the drug concentration, what is the concentration expected in the brain after drug intake orally? Given that all effects stated in the paper are in the brain tissue, the actual concentration in the brain must be considered and carefully discussed.
  5. There are essential missing controls in this study. The two other groups should be additionally included: wild-type animals treated with quercetin and those with rutin. The effects of the treatment in the mutant mouse line must be compared with the age-matched wild-type animals that received equal treatment.  
  6. The way how the authors presented their data looks odd. The data were normalised to the mutant mouse line in all figures and graphs. Why not normalise values to control groups as it has been typically used?
  7. Given that statistics include only 3 samples, demonstrating the data scatter is necessary to understand the variability between the experiments.
  8. Did the authors test all the data for normal distribution? Using ANOVA assumes parametric data – it must be clearly stated. 
  9. All axes should be clearly labelled, so readers perceive the data shown on the plots. Coding the X-axis makes it challenging for readers to comfortably pick up the experimental groups displayed on numerous plots. 
  10. The authors show only PCR and biochemical data, which are strong enough approaches but insufficient to claim the effects on brain redox homeostasis. Some additional evidence must be provided, at least immunohistology, redox proteomics, others. 
  11. The authors should describe all the data in the results section, even if they placed some data sets into the Supplementary Information. The first mention of some data placed into Supplements in the Discussion is incorrect.
  12.  Full names should be given before using their abbreviations at the first mention (numerously in the abstract), even if an abbreviation list is provided.

1The text needs rigorous editing for the style and language, avoiding odd word selections, and using articles and punctuation. Ideally, the authors have to enquire about the help of professional service.

Author Response

Dear and respected Reviewer-1,

We are thankful for your concern regarding the submission of our research manuscript. On behalf of all contributing authors, I would like to express our sincere appreciation for your letter with your constructive comments concerning our article entitled «A diet containing rutin ameliorates brain intracellular redox homeostasis in a mouse model of Alzheimer’s disease». These comments are all valuable and helpful for improving our article.

We will address all the queries you have raised. Please, see the attached document.

Reviewer 2 Report

In the manuscript ‘A diet containing rutin ameliorates brain intracellular redox homeostasis in a mouse model of Alzheimer’s disease’ authors explored the effects of quercetin and rutin on brain intracellular redox homeostasis (GSH/GSSG) linkage with BACE1 activity and APP expression in a transgenic TgAPP mouse (bearing human Swedish mutation APP transgene, APPswe).

Although the manuscript is well written, neat, and the material and methods well explained, in the reviewer's opinion, the manuscript has criticism.

Major comments:

-          The manuscript has not followed the guidelines of the journal, which shows little concern on the part of the authors. Bibliography, abstract, highlights, etc. have a different format than the one specific to the journal

-          The introduction section is brief and does not provide the background to the article. This should be widely improved.

-          In figures: the numbers 1,2,3 and 4 should be replaced by WT, TgAPP, TgAPP + Q and TgAPP + R.

-          I understand what the authors mean by placing the values of 1 (in the case of PCR) and 100% (in the rest of the tests). However, even if the mean of the group is made to obtain 1 or 100%, in the case of animals, the variability of the group must always be accounted for. In this sense (and as has been done in Figures 2, 3 and 4) bar 2 must contemplate group variability. All statistics have to be redone.

-          Why hasn't the % been made with respect to the control group of the activities of the antioxidant enzymes as has been done below for the rest of the techniques?

-          Below each figure it is specified that the statistics used were a Student-Newman-Keuls Test. However, section 2.10. Statistical analyzes specify that an ANOVA followed by the Newman-Keuls multiple comparison was carried out.

-          Furthermore, it is important to 1) specify the body weight variability of each group of animals and 2) specify how much food each group ate in these types of animal studies. This is because the observed effects may be due to variations in these aspects.

Minor comments:

-          In my opinion table 1,2,3 and 4 can go as supplementary material

-          Furthermore, it is unnecessary to put the formula for -ΔΔCt

Author Response

Dear and respected Reviewer-2,

On behalf of all contributing authors, I would like to express our sincere appreciation for your letter with your constructive comments concerning our article entitled «A diet containing rutin ameliorates brain intracellular redox homeostasis in a mouse model of Alzheimer’s disease». These comments are all valuable and helpful for improving our article.

We provide responses to the points you have raised. Please, see the attached document.

Round 2

Reviewer 1 Report

The authors have ignored all main issues listed in the previous reports, including even the data presentation, to show the variability of the results between all tested animals. In general, the conclusions are not supported by the results. 

Reviewer 2 Report

The authors must take into account the variability of the tgapp control group to perform the statistical analysis. Being mice, they must consider the variability within the group. Imagine that I want to analyze the expression of catalase in the brain of mice. Although finally with the ddct method the average is 1, you have mice with higher and lower ct. The same as in percentage, although the final percentage is 100%, there may be mice in 50 and others in 200%.

The bibliography is still not in the journal format.

If student-t was performed before and now ANOVA is performed. does it really not change any result?